# TOWARDS THE FIRST ADVERSARIALLY ROBUST NEURAL NETWORK MODEL ON MNIST

Lukas Schott[1,3*], Jonas Rauber[1,3*], Matthias Bethge[1,3,4†] & Wieland Brendel[1,3†]

[1]Centre for Integrative Neuroscience, University of Tübingen
[2]International Max Planck Research School for Intelligent Systems
[3]Bernstein Center for Computational Neuroscience Tübingen
[4]Max Planck Institute for Biological Cybernetics
[*]Joint first authors
[†]Joint senior authors
`firstname.lastname@bethgelab.org`

## ABSTRACT

Despite much effort, deep neural networks remain highly susceptible to tiny input perturbations and even for MNIST, one of the most common toy datasets in computer vision, no neural network model exists for which adversarial perturbations are large and make semantic sense to humans. We show that even the widely recognized and by far most successful $L_\infty$ defense by Madry et al. (1) has lower $L_0$ robustness than undefended networks and is still highly susceptible to $L_2$ perturbations, (2) classifies unrecognizable images with high certainty, (3) performs not much better than simple input binarization and (4) features adversarial perturbations that make little sense to humans. These results suggest that MNIST is far from being solved in terms of adversarial robustness. We present a novel robust classification model that performs *analysis by synthesis* using learned class-conditional data distributions. We derive bounds on the robustness and go to great length to empirically evaluate our model using maximally effective adversarial attacks by (a) applying decision-based, score-based, gradient-based and transfer-based attacks for several different $L_p$ norms, (b) by designing a new attack that exploits the structure of our defended model and (c) by devising a novel decision-based attack that seeks to minimize the number of perturbed pixels ($L_0$). The results suggest that our approach yields state-of-the-art robustness on MNIST against $L_0$, $L_2$ and $L_\infty$ perturbations and we demonstrate that most adversarial examples are strongly perturbed towards the perceptual boundary between the original and the adversarial class.

## 1 INTRODUCTION

Deep neural networks (DNNs) are strikingly susceptible to *minimal adversarial perturbations* (Szegedy et al., 2013), perturbations that are (almost) imperceptible to humans but which can switch the class prediction of DNNs to basically any desired target class.

One key problem in finding successful defenses is the difficulty of reliably evaluating model robustness. It has been shown time and again (Athalye et al., 2018; Athalye & Carlini, 2018; Brendel & Bethge, 2017) that basically all defenses previously proposed did not increase model robustness but prevented existing attacks from finding minimal adversarial examples, the most common reason being masking of the gradients on which most attacks rely. The few verifiable defenses can only guarantee robustness within a small linear regime around the data points (Hein & Andriushchenko, 2017; Raghunathan et al., 2018).

The only defense currently considered effective (Athalye et al., 2018) is a particular type of adversarial training (Madry et al., 2018). On MNIST, as of today this method is able to reach an accuracy of 88.79% for adversarial perturbations with an $L_\infty$ norm bounded by $\epsilon = 0.3$ (Zheng et al., 2018). In other words, if we allow an attacker to perturb the brightness of each pixel by up to 0.3 (range $[0, 1]$),

then he can only trick the model on $\approx 10\%$ of the samples. This is a great success, but does the model really learn more causal features to classify MNIST? We here demonstrate that this is not the case: For one, the defense by Madry et al. (SOTA on $L_\infty$) has lower $L_0$ robustness than undefended networks and is still highly susceptible in the $L_2$ metric. Second, the robustness results by Madry et al. can also be achieved with a simple input quantization because of the binary nature of single pixels in MNIST (which are typically either completely black or white) (Schmidt et al., 2018). Third, it is straight-forward to find unrecognizable images that are classified as a digit with high certainty. Finally, the minimum adversarial examples we find for the defense by Madry et al. make little to no sense to humans.

Taken together, even MNIST cannot be considered solved with respect to adversarial robustness. By "solved" we mean a model that reaches at least $99\%$ accuracy (see accuracy-vs-robustness trade-off (Tsipras et al., 2018; Bubeck et al., 2018)) and whose adversarial examples carry semantic meaning to humans (by which we mean that they start looking like samples that could belong to either class). Hence, despite the fact that MNIST is considered "too easy" by many and a mere toy example, finding adversarially robust models on MNIST is still an open problem.

A potential solution we explore in this paper is inspired by unrecognizable images (Nguyen et al., 2015) or *distal adversarials*. Distal adversarials are images that do not resemble images from the training set but which typically look like noise while still being classified by the model with high confidence. It seems difficult to prevent such images in feedforward networks as we have little control over how inputs are classified that are far outside of the training domain. In contrast, generative models can learn the distribution of their inputs and are thus able to gauge their confidence accordingly. By additionally learning the image distribution within each class we can check that the classification makes sense in terms of the image features being present in the input (e.g. an image of a bus should contain actual bus features). Following this line of thought from an information-theoretic perspective, one arrives at the well-known concept of Bayesian classifiers. We here introduce a fine-tuned variant based on variational autoencoders (Kingma & Welling, 2013) that combines robustness with high accuracy.

In summary, the contributions of this paper are as follows:

- We show that MNIST is unsolved from the point of adversarial robustness: the SOTA defense of Madry et al. (2018) is still highly vulnerable to tiny perturbations that are meaningless to humans.

- We introduce a new robust classification model and derive instance-specific robustness guarantees.

- We develop a strong attack that leverages the generative structure of our classification model.

- We introduce a novel decision-based attack that minimizes $L_0$.

- We perform an extensive evaluation of our defense across many attacks to show that it surpasses SOTA on $L_0$, $L_2$ and $L_\infty$ and features many adversarials that carry semantic meaning to humans.

We have evaluated the proposed defense to the best of our knowledge, but we are aware of the (currently unavoidable) limitations of evaluating robustness. We will release the model architecture and trained weights as a friendly invitation to fellow researchers to evaluate our model independently.

## 2 Related Work

The many defenses against adversarial attacks can roughly be subdivided into four categories:

- **Adversarial training:** The training data is augmented with adversarial examples to make models more robust (Madry et al., 2018; Szegedy et al., 2013; Tramèr et al., 2017; Ilyas et al., 2017).

- **Manifold projections:** An input sample is projected onto a learned data manifold (Samangouei et al., 2018; Ilyas et al., 2017; Shen et al., 2017; Song et al., 2018).

- **Stochasticity:** Certain inputs or hidden activations are shuffled or randomized (Prakash et al., 2018; Dhillon et al., 2018; Xie et al., 2018).

- **Preprocessing:** Inputs or hidden activations are quantized, projected into a different representation or are otherwise preprocessed (Buckman et al., 2018; Guo et al., 2018; Kabilan et al., 2018).

There has been much work showing that basically all defenses suggested so far in the literature do not substantially increase robustness over undefended neural networks (Athalye et al., 2018; Brendel &

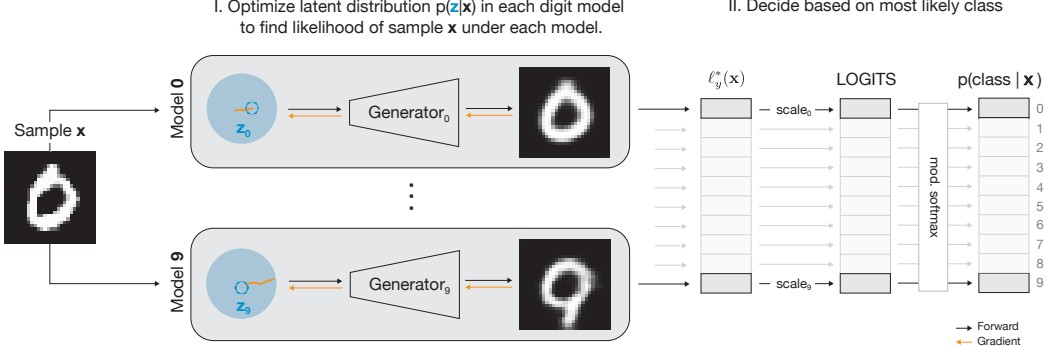

Figure 1: Overview over model architecture. In a nutshell: I) for each sample **x** we compute a lower bound on the log-likelihood (ELBO) under each class using gradient descent in the latent space. II) A class-dependent scalar weighting of the class-conditional ELBOs forms the final class prediction.

Bethge, 2017). The only widely accepted exception according to Athalye et al. (2018) is the defense by Madry et al. (2018) which is based on data augmentation with adversarials found by iterative projected gradient descent with random starting points. However, as we see in the results section, this defense is limited to the metric it is trained on ($L_\infty$) and it is straight-forward to generate small adversarial perturbations that carry little semantic meaning for humans.

Some other defenses have been based on generative models. Typically these defenses use the generative model to project onto the (learned) manifold of "natural" inputs. This includes in particular DefenseGAN (Samangouei et al., 2018), Adversarial Perturbation Elimination GAN (Shen et al., 2017) and Robust Manifold Defense (Ilyas et al., 2017), all of which project an image onto the manifold defined by a generator network $G$. The generated image is then classified by a discriminator in the usual way. A similar idea is used by PixelDefend (Song et al., 2018) which uses an autoregressive probabilistic method to learn the data manifold. Other ideas in similar directions include the use of denoising autoencoders (Liao et al., 2017) as well as MagNets (Meng & Chen, 2017), which projects or rejects inputs depending on their distance to the data manifold. All of these proposed defenses except for the defense by Ilyas et al. (2017) have been tested by Athalye et al. (2018); Athalye & Carlini (2018); Carlini & Wagner (2017) and others, and shown to be ineffective. It is straight-forward to understand why: For one, many adversarials still look like normal data points to humans. Second, the classifier on top of the projected image is as vulnerable to adversarial examples as before. Hence, for any data set with a natural amount of variation there will almost always be a certain perturbation against which the classifier is vulnerable and which can be induced by the right inputs.

We here follow a different approach by modeling the input distribution within each class (instead of modeling a single distribution for the complete data), and by classifying a new sample according to the class under which it has the highest likelihood. This approach, commonly referred to as a Bayesian classifier, gets away without any additional and vulnerable classifier. A very different but related approach is the work by George et al. (2017) which suggested a generative compositional model of digits to solve cluttered digit scenes like Captchas (adversarial robustness was not evaluated).

## 3    MODEL DESCRIPTION

Intuitively, we want to learn a causal model of the inputs (Schölkopf, 2017). Consider a cat: we want a model to learn that cats have four legs and two pointed ears, and then use this model to check whether a given input can be generated with these features. This intuition can be formalized as follows. Let $(\mathbf{x}, y)$ with $\mathbf{x} \in \mathbb{R}^N$ be an input-label datum. Instead of directly learning a posterior $p(y|\mathbf{x})$ from inputs to labels we now learn generative distributions $p(\mathbf{x}|y)$ and classify new inputs using Bayes formula,

$$p(y|\mathbf{x}) = \frac{p(\mathbf{x}|y)p(y)}{p(\mathbf{x})} \propto p(\mathbf{x}|y)p(y). \tag{1}$$

The label distribution $p(y)$ can be estimated from the training data. To learn the class-conditional sample distributions $p(\mathbf{x}|y)$ we use variational autoencoders (VAEs) (Kingma & Welling, 2013). VAEs estimate the log-likelihood $\log p(\mathbf{x})$ by learning a probabilistic generative model $p_\theta(\mathbf{x}|\mathbf{z})$

with latent variables $\mathbf{z} \sim p(\mathbf{z})$ and parameters $\theta$ (see Appendix A.3 for the full derivation). For class-conditional VAEs we can derive a lower bound on the log-likelihood $\log p(\mathbf{x}|y)$ as

$$\log p(\mathbf{x}|y) \geq \mathbb{E}_{\mathbf{z} \sim q_\phi(\mathbf{z}|\mathbf{x},y)} \left[ \log p_\theta(\mathbf{x}|\mathbf{z}, y) \right] - \mathcal{D}_{KL} \left[ q_\phi(\mathbf{z}|\mathbf{x}, y) || p(\mathbf{z}) \right] =: \ell_y(\mathbf{x}), \tag{2}$$

where $p(\mathbf{z}) = \mathcal{N}(\mathbf{0}, \mathbb{1})$ is a simple normal prior and $q_\phi(\mathbf{z}|\mathbf{x}, y)$ is the variational posterior with parameters $\phi$. The first term on the RHS is basically a reconstruction error while the second term on the RHS is the mismatch between the variational and the true posterior. The term on the RHS is the so-called evidence lower bound (ELBO) on the log-likelihood (Kingma & Welling, 2013). We implement the conditional distributions $p_\theta(\mathbf{x}|\mathbf{z}, y)$ and $q_\phi(\mathbf{z}|\mathbf{x}, y)$ as normal distributions for which the means are parametrized as DNNs (all details and hyperparameters are reported in Appendix A.7).

Our *Analysis by Synthesis* model (ABS) is illustrated in Figure 1. It combines several elements to simultaneously achieve high accuracy and robustness against adversarial perturbations:

- **Class-conditional distributions:** For each class $y$ we train a variational autoencoder $\text{VAE}_y$ on the samples of class $y$ to learn the class-conditional distribution $p(\mathbf{x}|y)$. This allows us to estimate a lower bound $\ell_y(\mathbf{x})$ on the log-likelihood of sample $\mathbf{x}$ under each class $y$.

- **Optimization-based inference:** The variational inference $q_\phi(\mathbf{z}|\mathbf{x}, y)$ is itself a neural network susceptible to adversarial perturbations. We therefore only use variational inference during training and perform "exact" inference over $p_\theta(\mathbf{x}|\mathbf{z}, y)$ during evaluation. This "exact" inference is implemented using gradient descent in the latent space (with fixed posterior width) to find the optimal $\mathbf{z}_y$ which maximizes the lower bound on the log-likelihood for each class:

$$\ell_y^*(\mathbf{x}) = \max_{\mathbf{z}} \ \log p_\theta(\mathbf{x}|\mathbf{z}, y) - \mathcal{D}_{KL} \left[ \mathcal{N}(\mathbf{z}, \sigma_q \mathbb{1}) || \mathcal{N}(\mathbf{0}, \mathbb{1}) \right]. \tag{3}$$

Note that we replaced the expectation in equation 2 with a maximum likelihood sample to avoid stochastic sampling and to simplify optimization. To avoid local minima we evaluate 8000 random points in the latent space of each VAE, from which we pick the best as a starting point for a gradient descent with 50 iterations using the Adam optimizer (Kingma & Ba, 2014).

- **Classification and confidence:** Finally, to perform the actual classification, we scale all $\ell_y^*(\mathbf{x})$ with a factor $\alpha$, exponentiate, add an offset $\eta$ and divide by the total evidence (like in a softmax),

$$p(y|\mathbf{x}) = \left( e^{\alpha \ell_y^*(\mathbf{x})} + \eta \right) / \sum_c \left( e^{\alpha \ell_c^*(\mathbf{x})} + \eta \right). \tag{4}$$

We introduced $\eta$ for the following reason: even on points far outside the data domain, where all likelihoods $q(\mathbf{x}, y) = e^{\alpha \ell_y^*(\mathbf{x})} + \eta$ are small, the standard softmax ($\eta = 0$) can lead to sharp posteriors $p(y|\mathbf{x})$ with high confidence scores for one class. This behavior is in stark contrast to humans, who would report a uniform distribution over classes for unrecognizable images. To model this behavior we set $\eta > 0$: in this case the posterior $p(y|\mathbf{x})$ converges to a uniform distribution whenever the maximum $q(\mathbf{x}, y)$ gets small relative to $\eta$. We chose $\eta$ such that the median confidence $p(y|\mathbf{x})$ is 0.9 for the predicted class on clean test samples. Furthermore, for a better comparison with cross-entropy trained networks, the scale $\alpha$ is trained to minimize the cross-entropy loss. We also tested this graded softmax in standard feedforward CNNs but did not find any improvement with respect to unrecognizable images.

- **Binarization (*Binary ABS* only):** The pixel intensities of MNIST images are almost binary. We exploit this by projecting the intensity $b$ of each pixel to 0 if $b < 0.5$ or 1 if $b \geq 0.5$ during testing.

- **Discriminative finetuning (*Binary ABS* only):** To improve the accuracy of the *Binary ABS* model we multiply $\ell_y^*(\mathbf{x})$ with an additional class-dependent scalar $\gamma_y$. The scalars are learned discriminatively (see A.7) and reach values in the range $\gamma_y \in [0.96, 1.06]$ for all classes $y$.

On important ingredient for the robustness of the ABS model is the Gaussian posterior in the reconstruction term which ensures that small changes in the input (in terms of L2) can only entail small changes to the posterior likelihood and thus to the model decision.

## 4 TIGHT ESTIMATES OF THE LOWER BOUND FOR ADVERSARIAL EXAMPLES

The decision of the model depends on the likelihood in each class, which for clean samples is mostly dominated by the posterior likelihood $p(\mathbf{x}|\mathbf{z})$. Because we chose this posterior to be Gaussian, the

class-conditional likelihoods can only change gracefully with changes in $\mathbf{x}$, a property which allows us to derive lower bounds on the model robustness. To see this, note that equation 3 can be written as,

$$\ell_c^*(\mathbf{x}) = \max_{\mathbf{z}} \; -\mathcal{D}_{KL}\left[\mathcal{N}(\mathbf{z}, \sigma_q \mathbb{1}) || \mathcal{N}(\mathbf{0}, \mathbb{1})\right] - \frac{1}{2\sigma^2} \|\mathbf{G}_c(\mathbf{z}) - \mathbf{x}\|_2^2 + C, \tag{5}$$

where we absorbed the normalization constants of $p(\mathbf{x}|\mathbf{z})$ into $C$ and $\mathbf{G}_c(\mathbf{z})$ is the mean of $p(\mathbf{x}|\mathbf{z}, c)$. Let $y$ be the ground-truth class and let $\mathbf{z}_\mathbf{x}^*$ be the optimal latent for the clean sample $\mathbf{x}$ for class $y$. We can then estimate a lower bound on $\ell_y^*(\mathbf{x} + \boldsymbol{\delta})$ for a perturbation $\boldsymbol{\delta}$ with size $\epsilon = \|\boldsymbol{\delta}\|_2$ (see derivation in Appendix A.4),

$$\ell_y^*(\mathbf{x} + \boldsymbol{\delta}) \geq \ell_y^*(\mathbf{x}) - \frac{1}{\sigma^2} \epsilon \|\mathbf{G}_y(\mathbf{z}_\mathbf{x}^*) - \mathbf{x}\|_2 - \frac{1}{2\sigma^2} \epsilon^2 + C. \tag{6}$$

Likewise, we can derive an upper bound of $\ell_y^*(\mathbf{x} + \boldsymbol{\delta})$ for all other classes $c \neq y$ (see Appendix A.5),

$$\ell_c^*(\mathbf{x} + \boldsymbol{\delta}) \leq -\mathcal{D}_{KL}\left[\mathcal{N}(\mathbf{0}, \sigma_q \mathbb{1}) || \mathcal{N}(\mathbf{0}, \mathbb{1})\right] + C - \begin{cases} \frac{1}{2\sigma^2}(d_c - \epsilon)^2 & \text{if } d_c \geq \epsilon \\ 0 & \text{else} \end{cases}. \tag{7}$$

for $d_c = \min_z \|\mathbf{G}_c(\mathbf{z}) - \mathbf{x}\|_2$. Now we can find $\epsilon$ for a given image $\mathbf{x}$ by equating $(7) = (6)$,

$$\epsilon_x = \min_{c \neq y} \max \left\{ 0, \frac{d_c + \ell_y^*(\mathbf{x}) - \mathcal{D}_{KL}\left[\mathcal{N}(\mathbf{0}, \sigma_q \mathbb{1}) || \mathcal{N}(\mathbf{0}, \mathbb{1})\right]}{2(d_c + \|\mathbf{G}_y(\mathbf{z}_\mathbf{x}^*) - \mathbf{x}\|_2)} \right\}. \tag{8}$$

Note that one assumption we make is that we can find the global minimum of $\|\mathbf{G}_c(\mathbf{z}) - \mathbf{x}\|_2^2$. In practice we generally find a very tight estimate of the global minimum (and thus the lower bound) because we optimize in a smooth and low-dimensional space and because we perform an additional brute-force sampling step. We provide quantitative values for $\epsilon$ in section 7.

## 5 Adversarial Attacks

Reliably evaluating model robustness is difficult because each attack only provides an upper bound on the size of the adversarial perturbations (Uesato et al., 2018). To make this bound as tight as possible we apply many different attacks and choose the best one for each sample and model combination (using the implementations in Foolbox v1.3 (Rauber et al., 2017) which often perform internal hyperparameter optimization). We also created a novel decision-based $L_0$ attack as well as a customized attack that specifically exploits the structure of our model. Nevertheless, we cannot rule out that more effective attacks exist and we will release the trained model for future testing.

**Latent Descent attack** This novel attack exploits the structure of the ABS model. Let $\mathbf{x}_t$ be the perturbed sample $\mathbf{x}$ in iteration $t$. We perform variational inference $p(\mathbf{z}|\mathbf{x}_t, y) = \mathcal{N}(\boldsymbol{\mu}_y(\mathbf{x}_t), \sigma_q \boldsymbol{I})$ to find the most likely class $\tilde{y}$ that is different from the ground-truth class. We then make a step towards the maximum likelihood posterior $p(\mathbf{x}|\mathbf{z}, \tilde{y})$ of that class which we denote as $\tilde{\mathbf{x}}_{\tilde{y}}$,

$$\mathbf{x}_t \mapsto (1 - \epsilon)\mathbf{x}_t + \epsilon \tilde{\mathbf{x}}_{\tilde{y}}. \tag{9}$$

We choose $\epsilon = 10^{-2}$ and iterate until we find an adversarial. For a more precise estimate we perform a subsequent binary search of 10 steps within the last $\epsilon$ interval. Finally, we perform another binary search between the adversarial and the original image to reduce the perturbation as much as possible.

**Decision-based attacks** We use several decision-based attacks because they do not rely on gradient information and are thus insensitive to gradient masking or missing gradients. In particular, we apply the *Boundary Attack* (Brendel et al., 2018), which is competitive with gradient-based attacks in minimizing the $L_2$ norm, and introduce the *Pointwise Attack*, a novel decision-based attack that greedily minimizes the $L_0$ norm. It first adds salt-and-pepper noise until the image is misclassified and then repeatedly iterates over all perturbed pixels, resetting them to the clean image if the perturbed image stays adversarial. The attack ends when no pixel can be reset anymore. We provide an implementation of the attack in Foolbox (Rauber et al., 2017). Finally, we apply two simple noise attacks, the *Gaussian Noise* attack and the *Salt&Pepper Noise* attack as baselines.

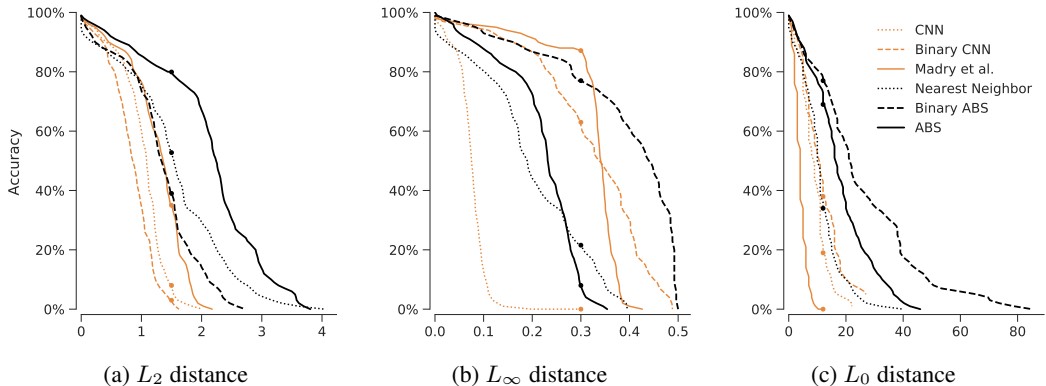

Figure 2: Accuracy-distortion plots for each distance metric and all models. In (b) we see that a threshold at 0.3 favors Madry et al. while a threshold of 0.35 would have favored the Binary ABS.

**Transfer-based attacks** Transfer attacks also don't rely on gradients of the target model but instead compute them on a substitute: given an input $\mathbf{x}$ we first compute adversarial perturbations $\boldsymbol{\delta}$ on the substitute using different gradient-based attacks ($L_2$ and $L_\infty$ Basic Iterative Method (BIM), Fast Gradient Sign Method (FGSM) and $L_2$ Fast Gradient Method) and then perform a line search to find the smallest $\epsilon$ for which $\mathbf{x} + \epsilon\boldsymbol{\delta}$ (clipped to the range $[0, 1]$) is still an adversarial for the target model.

**Gradient-based attacks** We apply the Momentum Iterative Method (MIM) (Dong et al., 2017) that won the NIPS 2017 adversarial attack challenge, the Basic Iterative Method (BIM) (Kurakin et al., 2016) (also known as Projected Gradient Descent (PGD))—for both the $L_2$ and the $L_\infty$ norm—as well as the Fast Gradient Sign Method (FGSM) (Goodfellow et al., 2014) and its $L_2$ variant, the Fast Gradient Method (FGM). For models with input binarization (Binary CNN, Binary ABS), we obtain gradients using the straight-through estimator (Bengio et al., 2013).

**Score-based attacks** We additionally run all attacks listed under *Gradient-based attacks* using numerically estimated gradients (possible for all models). We use a simple coordinate-wise finite difference method (NES estimates (Ilyas et al., 2018) performed comparable or worse) and repeat the attacks with different values for the step size of the gradient estimator.

**Postprocessing (binary models only)** For models with input binarization (sec. 6) we postprocess all adversarials by setting pixel intensities either to the corresponding value of the clean image or the binarization threshold (0.5). This reduces the perturbation size without changing model decisions.

## 6 EXPERIMENTS

We compare our ABS model as well as two ablations—ABS with input binarization during test time (Binary ABS) and a CNN with input binarization during train and test time (Binary CNN)—against three other models: the SOTA $L_\infty$ defense (Madry et al., 2018)[1], a Nearest Neighbour (NN) model (as a somewhat robust but not accurate baseline) and a vanilla CNN (as an accurate but not robust baseline), see Appendix A.7. We run all attacks (see sec. 5) against all applicable models.

For each model and $L_p$ norm, we show how the accuracy of the models decreases with increasing adversarial perturbation size (Figure 2) and report two metrics: the median adversarial distance (Table 1, left values) and the model's accuracy against bounded adversarial perturbations (Table 1, right values). The median of the perturbation sizes (Table 1, left values) is robust to outliers and summarizes most of the distributions quite well. It represents the perturbation size for which the particular model achieves 50% accuracy and does not require the choice of a threshold. Clean samples that are already misclassified are counted as adversarials with a perturbation size equal to 0, failed attacks as $\infty$. The commonly reported model accuracy on bounded adversarial perturbations, on the other hand, requires a metric-specific threshold that can bias the results. We still report it (Table 1, right values) for completeness and set $\epsilon_{L_2} = 1.5$, $\epsilon_{L_\infty} = 0.3$ and $\epsilon_{L_0} = 12$ as thresholds.

---

[1]We used the trained model provided by the authors: https://github.com/MadryLab/mnist_challenge

| | CNN | Binary CNN | Nearest Neighbor | Madry et al. | Binary ABS | ABS |
|---|---|---|---|---|---|---|
| Clean | 99.1% | 98.5% | 96.9% | 98.8% | 99.0% | 99.0% |
| $L_2$-metric ($\epsilon = 1.5$) | | | | | | |
| Transfer Attacks | 1.1 / 14% | 1.4 / 38% | 5.4 / 90% | 3.7 / 94% | 2.5 / 86% | 4.6 / 94% |
| Gaussian Noise | 5.2 / 96% | 3.4 / 92% | $\infty$ / 91% | 5.4 / 96% | 5.6 / 89% | 10.9 / 98% |
| Boundary Attack | 1.2 / 21% | 3.3 / 84% | 2.9 / 73% | 1.4 / 37% | 6.0 / 91% | 2.6 / 83% |
| Pointwise Attack | 3.4 / 91% | 1.9 / 71% | 3.5 / 89% | 1.9 / 71% | 3.1 / 86% | 4.6 / 94% |
| FGM | 1.4 / 48% | 1.4 / 50% | | $\infty$ / 96% | | |
| FGM w/ GE | 1.4 / 42% | 2.8 / 51% | 3.7 / 79% | $\infty$ / 88% | 1.9 / 68% | 3.5 / 89% |
| DeepFool | 1.2 / 18% | 1.0 / 11% | | 9.0 / 91% | | |
| DeepFool w/ GE | 1.3 / 30% | 0.9 / 5% | 1.6 / 55% | 5.1 / 90% | 1.4 / 41% | 2.4 / 83% |
| L2 BIM | 1.1 / 13% | 1.0 / 11% | | 4.8 / 88% | | |
| L2 BIM w/ GE | 1.1 / 37% | $\infty$ / 50% | 1.7 / 62% | 3.4 / 88% | 1.6 / 63% | 3.1 / 87% |
| Latent Descent Attack | | | | | 2.6 / 97% | 2.7 / 85% |
| **All $L_2$ Attacks** | 1.1 / 8% | 0.9 / 3% | 1.5 / 53% | 1.4 / 35% | 1.3 / 39% | **2.3** / 80% |
| $L_\infty$-metric ($\epsilon = 0.3$) | | | | | | |
| Transfer Attacks | 0.08 / 0% | 0.44 / 85% | 0.42 / 78% | 0.39 / 92% | 0.49 / 88% | 0.34 / 73% |
| FGSM | 0.10 / 4% | 0.43 / 77% | | 0.45 / 93% | | |
| FGSM w/ GE | 0.10 / 21% | 0.42 / 71% | 0.38 / 68% | 0.47 / 89% | 0.49 / 85% | 0.27 / 34% |
| $L_\infty$ DeepFool | 0.08 / 0% | 0.38 / 74% | | 0.42 / 90% | | |
| $L_\infty$ DeepFool w/ GE | 0.09 / 0% | 0.37 / 67% | 0.21 / 26% | 0.53 / 90% | 0.46 / 78% | 0.27 / 39% |
| BIM | 0.08 / 0% | 0.36 / 70% | | 0.36 / 90% | | |
| BIM w/ GE | 0.08 / 37% | $\infty$ / 70% | 0.25 / 43% | 0.46 / 89% | 0.49 / 86% | 0.25 / 13% |
| MIM | 0.08 / 0% | 0.37 / 71% | | 0.34 / 90% | | |
| MIM w/ GE | 0.09 / 36% | $\infty$ / 69% | 0.19 / 26% | 0.36 / 89% | 0.46 / 85% | 0.26 / 17% |
| **All $L_\infty$ Attacks** | 0.08 / 0% | 0.34 / 64% | 0.19 / 22% | 0.34 / 88% | **0.44** / 77% | 0.23 / 8% |
| $L_0$-metric ($\epsilon = 12$) | | | | | | |
| Salt&Pepper Noise | 44.0 / 91% | 44.0 / 88% | 161.0 / 88% | 13.5 / 56% | 146.0 / 94% | 165.0 / 94% |
| Pointwise Attack 10x | 9.0 / 19% | 11.0 / 39% | 10.0 / 34% | 4.0 / 0% | 22.0 / 77% | 16.5 / 69% |
| **All $L_0$ Attacks** | 9.0 / 19% | 11.0 / 38% | 10.0 / 34% | 4.0 / 0% | **21.5** / 77% | 16.5 / 69% |

Table 1: Results for different models, adversarial attacks and distance metrics. Each entry shows the median adversarial distance across all samples (left value, black) as well as the model's accuracy against adversarial perturbations bounded by the thresholds $\epsilon_{L_2} = 1.5$, $\epsilon_{L_\infty} = 0.3$ and $\epsilon_{L_0} = 12$ (right value, gray). *"w/ GE"* indicates attacks that use numerical gradient estimation.

## 7 RESULTS

**Minimal Adversarials** Our robustness evaluation results of all models are reported in Table 1 and Figure 2. All models except the Nearest Neighbour classifier perform close to 99% accuracy on clean test samples. We report results for three different norms: $L_2$, $L_\infty$ and $L_0$.

- For $L_2$ our ABS model outperforms all other models by a large margin.
- For $L_\infty$, our Binary ABS model is state-of-the-art in terms of median perturbation size. In terms of accuracy (perturbations $< 0.3$), Madry et al. seems more robust. However, as revealed by the accuracy-distortion curves in Figure 2, this is an artifact of the specific threshold (Madry et al. is optimized for $0.3$). A slightly larger one (e.g. $0.35$) would strongly favor the Binary ABS model.
- For $L_0$, both ABS and Binary ABS are much more robust than all other models. Interestingly, the model by Madry et al. is the least robust, even less than the baseline CNN.

In Figure 3 we show adversarial examples. For each sample we show the minimally perturbed $L_2$ adversarial found by any attack. Adversarials for the baseline CNN and the Binary CNN are almost

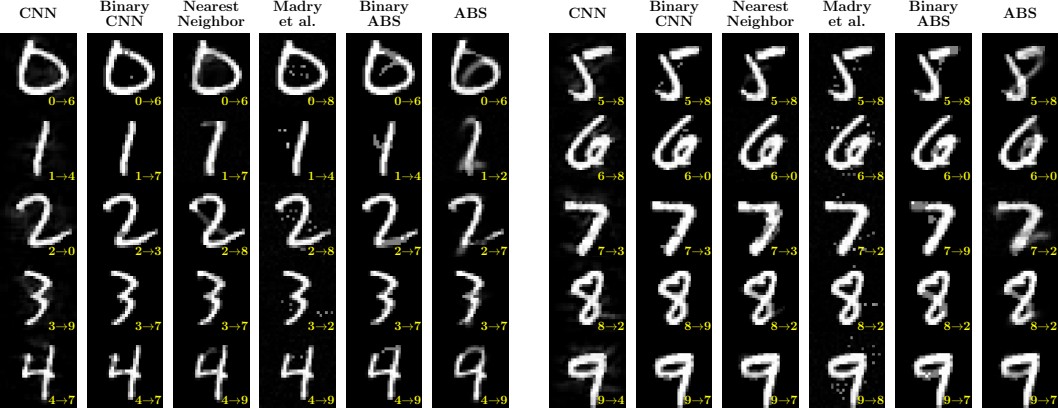

Figure 3: Adversarial examples for the ABS models are perceptually meaningful: For each sample (randomly chosen from each class) we show the minimally perturbed $L_2$ adversarial found by any attack. Our ABS models have clearly visible and often semantically meaningful adversarials. Madry et al. requires perturbations that are clearly visible, but their semantics are less clear.

imperceptible. The Nearest Neighbour model, almost by design, exposes (some) adversarials that interpolate between two numbers. The model by Madry et al. requires perturbations that are clearly visible but make little semantic sense to humans. Finally, adversarials generated for the ABS models are semantically meaningful for humans and are sitting close to the perceptual boundary between the original and the adversarial class. For a more thorough comparison see appendix Figures 5, 6 and 7.

**Lower bounds on Robustness** For the ABS models and the $L_2$ metric we estimate a lower bound of the robustness. The lower bound for the mean perturbation[2] for the MNIST test set is $\epsilon = 0.690 \pm 0.005$ for the ABS and $\epsilon = 0.601 \pm 0.005$ for the binary ABS. We estimated the error by using different random seeds for our optimization procedure and standard error propagation over 10 runs. With adversarial training Hein & Andriushchenko (2017) achieve a mean $L_2$ robustness guarantee of $\epsilon = 0.48$ while reaching 99% accuracy. In the $L_{inf}$ metric we find a median robustness of 0.06.

**Distal Adversarials** We probe the behavior of CNN, Madry et al. and our ABS model outside the data distribution. We start from random noise images and perform gradient ascent to maximize the output probability of a fixed label until $p(y|\mathbf{x}) \geq 0.9$ (as computed by the modified softmax from equation (8)). The results are visualized in Figure 4. Standard CNNs and Madry et al. provide high confidence class probabilities for unrecognizable images. Our ABS model does not provide high confidence predictions in out-of-distribution regions.

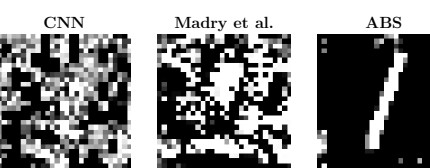

Figure 4: Images of ones classified with a probability above 90%.

## 8 DISCUSSION & CONCLUSION

In this paper we demonstrated that, despite years of work, we as a community failed to create neural networks that can be considered robust on MNIST from the point of human perception. In particular, we showed that even today's best defense is susceptible to small adversarial perturbations that make little to no semantic sense to humans. We presented a new approach based on *analysis by synthesis* that seeks to explain its inference by means of the actual image features. We performed an extensive analysis to show that minimal adversarial perturbations in this model are large across all tested $L_p$ norms and semantically meaningful to humans. Note that our architecture derives its robustness from its design and does not require any additionally training with adversarial examples.

We acknowledge that it is not easy to reliably evaluate a model's adversarial robustness and most defenses proposed in the literature have later been shown to be ineffective. In particular, the structure

---

[2]The mean instead of the median is reported to allow for a comparison with (Hein & Andriushchenko, 2017).

of the ABS model prevents the computation of gradients which might give the model an unfair advantage. We put a lot of effort into an extensive evaluation of adversarial robustness using a large collection of powerful attacks, including one specifically designed to be particularly effective against the ABS model (the *Latent Descent* attack), and we will release the model architecture and trained weights as a friendly invitation to fellow researchers to evaluate our model.

Looking at the results of individual attacks (Table 1) we find that there is no single attack that works best on all models, thus highlighting the importance for a broad range of attacks. Without the Boundary Attack, for example, Madry et al. would have looked more robust to $L_2$ adversarials than it is. For similar reasons Figure 6b of Madry et al. (2018) reports a median $L_2$ perturbation size larger than 5, compared to the 1.4 achieved by the Boundary Attack. Moreover,the combination of all attacks of one metric (*All $L_2$ / $L_\infty$ / $L_0$ Attacks*) is often better than any individual attack, indicating that different attacks are optimal on different samples.

Our conceptual implementation of the ABS model with one VAE per class neither scales efficiently to more classes nor to more complex datasets (a preliminary experiment on CIFAR10 provided only 54% test accuracy). However, first experiments on two class CIFAR indicate that the proposed model is also robust on CIFAR (we reach a median L2 robustness of 2.6 compared to 0.8 for a vanilla CNN, see Appendix A.1) for details). To increase the accuracy, there are many ways in which the ABS model can be improved, ranging from better and faster generative models (e.g. flow-based) to better training procedures.

In a nutshell, we demonstrated that MNIST is still not solved from the point of adversarial robustness and showed that our novel approach based on analysis by synthesis has great potential to reduce the vulnerability against adversarial attacks and to align machine perception with human perception.

### ACKNOWLEDGMENTS

This work has been funded, in part, by the German Federal Ministry of Education and Research (BMBF) through the Bernstein Computational Neuroscience Program Tübingen (FKZ: 01GQ1002) as well as the German Research Foundation (DFG CRC 1233 on "Robust Vision"). The authors thank the International Max Planck Research School for Intelligent Systems (IMPRS-IS) for supporting L.S. and J.R.; J.R. acknowledges support by the Bosch Forschungsstiftung (Stifterverband, T113/30057/17); W.B. was supported by the Carl Zeiss Foundation (0563-2.8/558/3); M.B. acknowledges support by the Centre for Integrative Neuroscience Tübingen (EXC 307); W.B. and M.B. were supported by the Intelligence Advanced Research Projects Activity (IARPA) via Department of Interior / Interior Business Center (DoI/IBC) contract number D16PC00003.

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

# A    APPENDIX

## A.1    TWO CLASS CIFAR

We estimate the robustness of our ABS model on two class CIFAR (airplane vs. automobile). Preliminary results suggest that our robustness is not limited to MNIST.

In order to adapt to CIFAR, we modified the ABS slightly by modifying encoder and decoder to fit (32x32x3) CIFAR images. We also increased the number of dimensions in the latent space form 8 to 20.

| Model | CNN | ABS |
|---|---|---|
| Accuracy | 97.1% | 89.7% |
| Median $L_2$ distance | 0.8 (with BIM) | 2.5 (with Latent Descent attack) |

Table 2: Accuracy and estimated robustness on two class CIFAR.

## A.2    FIGURES

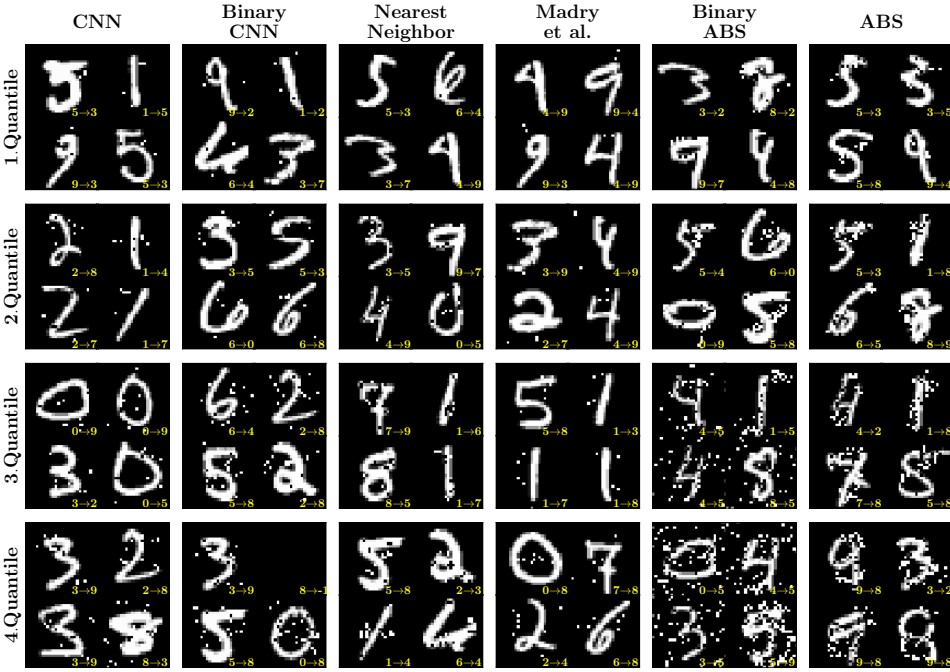

Figure 5: $L_0$ error quantiles: We always choose the minimally perturbed $L_0$ adversarial found by any attack for each model. For an unbiased selection, we then randomly sample images within four error quantiles ($0 - 25\%$, $25 - 50\%$, $50 - 75\%$, and $75 - 100\%$). Where $100\%$ corresponds to the maximal (over samples) minimum (over attacks) perturbation found for each model.

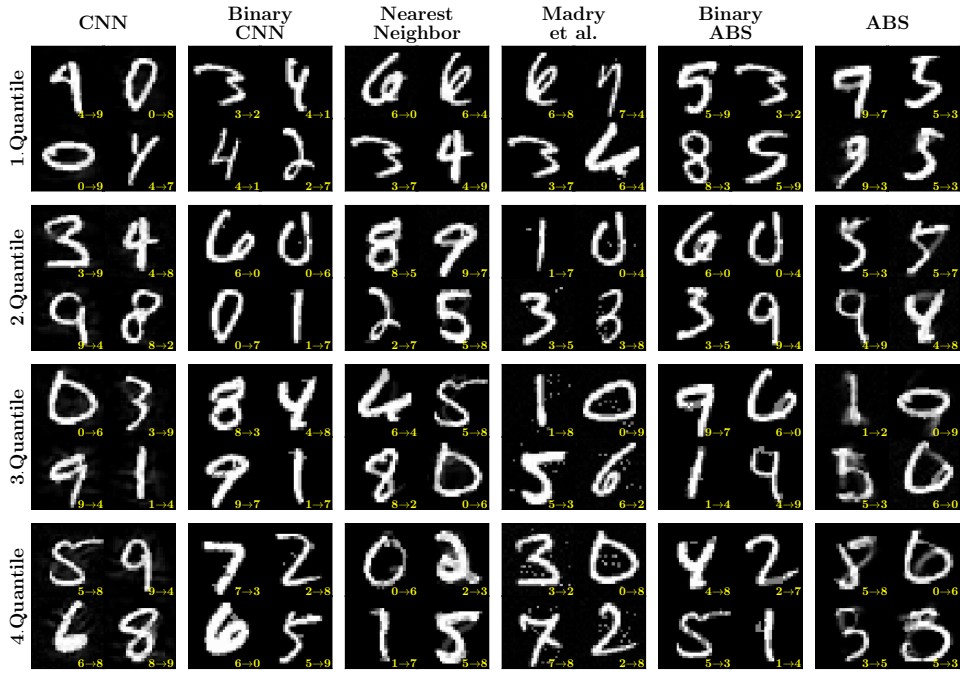

Figure 6: $L_2$ error quantiles: We always choose the minimally perturbed $L_2$ adversarial found by any attack for each model. For an unbiased selection, we then randomly sample 4 images within four error quantiles ($0 - 25\%$, $25 - 50\%$, $50 - 75\%$, and $75 - 100\%$).

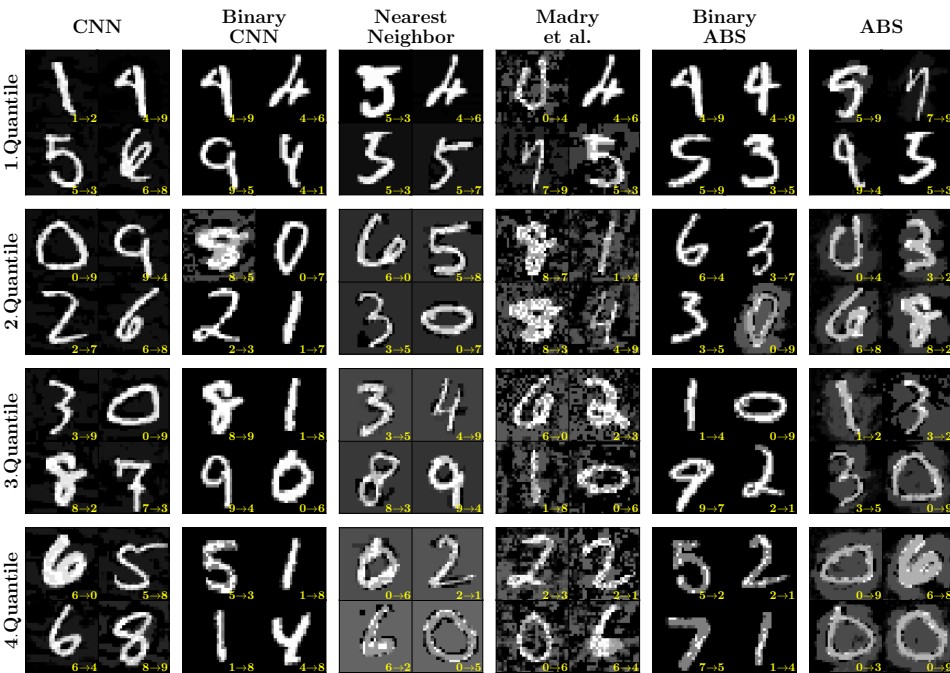

Figure 7: $L_\infty$ error quantiles: We always choose the minimally perturbed $L_\infty$ adversarial found by any attack for each model. For an unbiased selection, we then randomly sample images within four error quantiles ($0 - 25\%$, $25 - 50\%$, $50 - 75\%$, and $75 - 100\%$).

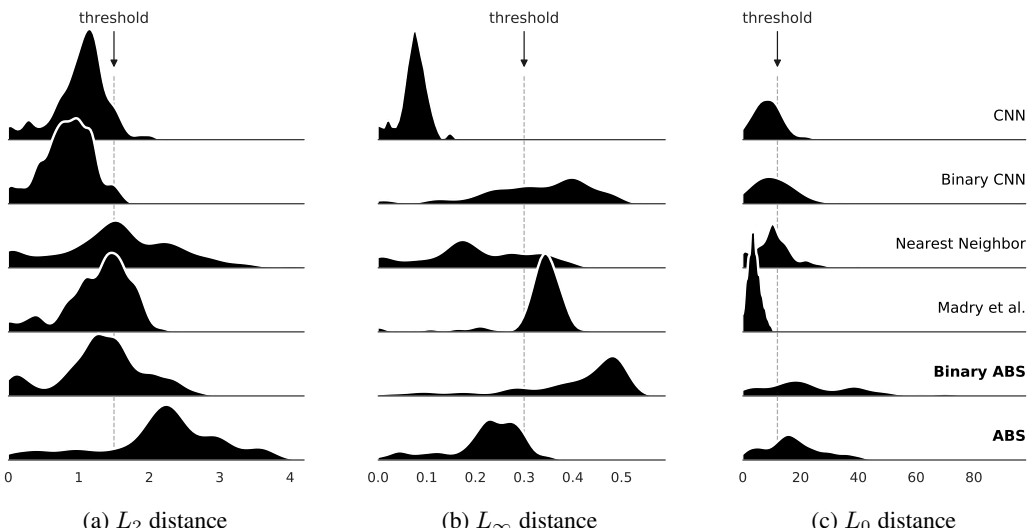

(a) $L_2$ distance      (b) $L_\infty$ distance      (c) $L_0$ distance

Figure 8: Distribution of minimal adversarials for each model and distance metric. In (b) we see that a threshold at $0.3$ favors Madry et al. while a threshold of $0.35$ would have favored the Binary ABS.

## A.3 DERIVATION I

Derivation of the ELBO in equation 2.

$$\log p_\theta(\mathbf{x}) = \log \int \mathbf{dz}\, p_\theta(\mathbf{x}|\mathbf{z})p(\mathbf{z}),$$

where $p(\mathbf{z}) = \mathcal{N}(\mathbf{0}, \mathbb{1})$ is a simple normal prior. Based on the idea of importance sampling using a variational posterior $q_\phi(\mathbf{z}|\mathbf{x})$ with parameters $\phi$ and using Jensen's inequality we arrive at

$$
\begin{aligned}
&= \log \int \mathbf{dz}\, \frac{q_\phi(\mathbf{z}|\mathbf{x})}{q_\phi(\mathbf{z}|\mathbf{x})} p_\theta(\mathbf{x}|\mathbf{z})p(\mathbf{z}), \\
&= \log \mathbb{E}_{\mathbf{z}\sim q_\phi(\mathbf{z}|\mathbf{x})} \left[ \frac{p_\theta(\mathbf{x}|\mathbf{z})p(\mathbf{z})}{q_\phi(\mathbf{z}|\mathbf{x})} \right], \\
&\geq \mathbb{E}_{\mathbf{z}\sim q_\phi(\mathbf{z}|\mathbf{x})} \left[ \log \frac{p_\theta(\mathbf{x}|\mathbf{z})p(\mathbf{z})}{q_\phi(\mathbf{z}|\mathbf{x})} \right], \\
&= \mathbb{E}_{\mathbf{z}\sim q_\phi(\mathbf{z}|\mathbf{x})} \left[ \log p_\theta(\mathbf{x}|\mathbf{z}) + \log \frac{p(\mathbf{z})}{q_\phi(\mathbf{z}|\mathbf{x})} \right], \\
&= \mathbb{E}_{\mathbf{z}\sim q_\phi(\mathbf{z}|\mathbf{x})} \left[ \log p_\theta(\mathbf{x}|\mathbf{z}) \right] - \mathcal{D}_{KL} \left[ q_\phi(\mathbf{z}|\mathbf{x}) || p(\mathbf{z}) \right].
\end{aligned}
$$

This lower bound is commonly referred to as ELBO.

## A.4 DERIVATION II: LOWER BOUND FOR $L_2$ ROBUSTNESS ESTIMATION

Derivation of equation 6. Starting from equation 3 we find that for a perturbation $\boldsymbol{\delta}$ with size $\epsilon = \|\boldsymbol{\delta}\|_2$ of sample $\mathbf{x}$ the lower bound $\ell_y^*(\mathbf{x} + \boldsymbol{\delta})$ can itself be bounded by,

$$
\begin{aligned}
\ell_y^*(\mathbf{x} + \boldsymbol{\delta}) &= \max_{\mathbf{z}} -\mathcal{D}_{KL}\left[\mathcal{N}(\mathbf{z}, \sigma_q \mathbb{1}) || \mathcal{N}(\mathbf{0}, \mathbb{1})\right] - \frac{1}{2\sigma^2} \|\mathbf{G}_y(\mathbf{z}) - \mathbf{x} - \boldsymbol{\delta}\|_2^2 + C, \\
&\geq -\mathcal{D}_{KL}\left[\mathcal{N}(\mathbf{z}_\mathbf{x}^*, \sigma_q \mathbb{1}) || \mathcal{N}(\mathbf{0}, \mathbb{1})\right] - \frac{1}{2\sigma^2} \|\mathbf{G}_y(\mathbf{z}_\mathbf{x}^*) - \mathbf{x} - \boldsymbol{\delta}\|_2^2 + C,
\end{aligned}
$$

where $\mathbf{z}_\mathbf{x}^*$ is the optimal latent vector for the clean sample $\mathbf{x}$ for class $y$,

$$
\begin{aligned}
&= \ell_y^*(\mathbf{x}) + \frac{1}{\sigma^2} \boldsymbol{\delta}^\top \left(\mathbf{G}_y(\mathbf{z}_\mathbf{x}^*) - \mathbf{x}\right) - \frac{1}{2\sigma^2}\epsilon^2 + C, \\
&\geq \ell_y^*(\mathbf{x}) - \frac{1}{\sigma^2}\epsilon \|\mathbf{G}_y(\mathbf{z}_\mathbf{x}^*) - \mathbf{x}\|_2 - \frac{1}{2\sigma^2}\epsilon^2 + C.
\end{aligned}
\tag{10}
$$

## A.5 DERIVATION III: UPPER BOUND FOR $L_2$ ROBUSTNESS ESTIMATION

Derivation of equation 7.

$$
\begin{aligned}
\ell_c^*(\mathbf{x} + \boldsymbol{\delta}) &= \max_{\mathbf{z}} -\mathcal{D}_{KL}\left[\mathcal{N}(\mathbf{z}, \sigma_q \mathbb{1}) || \mathcal{N}(\mathbf{0}, \mathbb{1})\right] - \frac{1}{2\sigma^2} \|\mathbf{G}_y(\mathbf{z}) - \mathbf{x} - \boldsymbol{\delta}\|_2^2 + C, \\
&\leq -\mathcal{D}_{KL}\left[\mathcal{N}(\mathbf{0}, \sigma_q \mathbb{1}) || \mathcal{N}(\mathbf{0}, \mathbb{1})\right] + C - \min_{\mathbf{z}} \frac{1}{2\sigma^2} \|\mathbf{G}_c(\mathbf{z}) - \mathbf{x} - \boldsymbol{\delta}\|_2^2, \\
&\leq -\mathcal{D}_{KL}\left[\mathcal{N}(\mathbf{0}, \sigma_q \mathbb{1}) || \mathcal{N}(\mathbf{0}, \mathbb{1})\right] + C - \min_{\mathbf{z}, \boldsymbol{\delta}} \frac{1}{2\sigma^2} \|\mathbf{G}_c(\mathbf{z}) - \mathbf{x} - \boldsymbol{\delta}\|_2^2, \\
&= -\mathcal{D}_{KL}\left[\mathcal{N}(\mathbf{0}, \sigma_q \mathbb{1}) || \mathcal{N}(\mathbf{0}, \mathbb{1})\right] + C - \begin{cases} \frac{1}{2\sigma^2}(d_c - \epsilon)^2 & \text{if } d_c \geq \epsilon \\ 0 & \text{else} \end{cases}.
\end{aligned}
\tag{11}
$$

for $d_c = \min_{\mathbf{z}} \|\mathbf{G}_c(\mathbf{z}) - \mathbf{x}\|_2$. The last equation comes from the solution of the constrained optimization problem $\min_d (d - \epsilon)^2 d$ s.t. $d > d_c$. Note that a tighter bound might be achieved by assuming single $\boldsymbol{\delta}$ for upper and lower bound.

## A.6 $L_\infty$ ROBUSTNESS ESTIMATION

We proceed in the same way as for $L_2$. Starting again from

$$
\ell_c^*(\mathbf{x}) = \max_{\mathbf{z}} -\mathcal{D}_{KL}\left[\mathcal{N}(\mathbf{z}, \sigma_q \mathbb{1}) || \mathcal{N}(\mathbf{0}, \mathbb{1})\right] - \frac{1}{2\sigma^2} \|\mathbf{G}_c(\mathbf{z}) - \mathbf{x}\|_2^2 + C,
\tag{12}
$$

let $y$ be the predicted class and let $\mathbf{z}_\mathbf{x}^*$ be the optimal latent for the clean sample $\mathbf{x}$ for class $y$. We can then estimate a lower bound on $\ell_y^*(\mathbf{x} + \boldsymbol{\delta})$ for a perturbation $\boldsymbol{\delta}$ with size $\epsilon = \|\boldsymbol{\delta}\|_\infty$,

$$\ell_y^*(\mathbf{x} + \boldsymbol{\delta}) = \max_\mathbf{z} -\mathcal{D}_{KL}\left[\mathcal{N}(\mathbf{z}, \sigma_q \mathbb{1})||\mathcal{N}(\mathbf{0}, \mathbb{1})\right] - \frac{1}{2\sigma^2}\|\mathbf{G}_y(\mathbf{z}) - \mathbf{x} - \boldsymbol{\delta}\|_2^2 + C,$$

$$\geq -\mathcal{D}_{KL}\left[\mathcal{N}(\mathbf{z}_\mathbf{x}^*, \sigma_q \mathbb{1})||\mathcal{N}(\mathbf{0}, \mathbb{1})\right] - \frac{1}{2\sigma^2}\|\mathbf{G}_y(\mathbf{z}_\mathbf{x}^*) - \mathbf{x} - \boldsymbol{\delta}\|_2^2 + C,$$

where $\mathbf{z}_\mathbf{x}^*$ is the optimal latent for the clean sample $\mathbf{x}$ for class $y$.

$$= \ell_y^*(\mathbf{x}) + \frac{1}{\sigma^2}\boldsymbol{\delta}^\top(\mathbf{G}_y(\mathbf{z}_\mathbf{x}^*) - \mathbf{x}) - \frac{1}{2\sigma^2}\|\boldsymbol{\delta}\|_2^2 + C,$$

$$\geq \ell_y^*(\mathbf{x}) + C + \frac{1}{2\sigma^2}\min_{\boldsymbol{\delta}}\left(2\boldsymbol{\delta}^\top(\mathbf{G}_y(\mathbf{z}_\mathbf{x}^*) - \mathbf{x}) - \|\boldsymbol{\delta}\|_2^2\right),$$

$$= \ell_y^*(\mathbf{x}) + C + \frac{1}{2\sigma^2}\sum_i \min_{\delta_i}\left(2\delta_i[\mathbf{G}_y(\mathbf{z}_\mathbf{x}^*) - \mathbf{x}]_i - \delta_i^2\right),$$

$$= \ell_y^*(\mathbf{x}) + C + \frac{1}{2\sigma^2}\sum_i \begin{cases} [\mathbf{G}_y(\mathbf{z}_\mathbf{x}^*) - \mathbf{x}]_i^2 & \text{if } |[\mathbf{G}_y(\mathbf{z}_\mathbf{x}^*) - \mathbf{x}]_i| \leq \epsilon \\ \epsilon \, |[\mathbf{G}_y(\mathbf{z}_\mathbf{x}^*) - \mathbf{x}]_i| & \text{else} \end{cases}. \tag{13}$$

Similarly, we can estimate an upper bound on $\ell_c^*(\mathbf{x} + \boldsymbol{\delta})$ on all other classes $c \neq y$,

$$\ell_c^*(\mathbf{x} + \boldsymbol{\delta}) \leq -\mathcal{D}_{KL}\left[\mathcal{N}(\mathbf{0}, \sigma_q \mathbb{1})||\mathcal{N}(\mathbf{0}, \mathbb{1})\right] + C - \min_\mathbf{z} \frac{1}{2\sigma^2}\|\mathbf{G}_c(\mathbf{z}) - \mathbf{x} - \boldsymbol{\delta}\|_2^2,$$

$$\leq -\mathcal{D}_{KL}\left[\mathcal{N}(\mathbf{0}, \sigma_q \mathbb{1})||\mathcal{N}(\mathbf{0}, \mathbb{1})\right] + C - \min_{\mathbf{z},\boldsymbol{\delta}} \frac{1}{2\sigma^2}\|\mathbf{G}_c(\mathbf{z}) - \mathbf{x} - \boldsymbol{\delta}\|_2^2,$$

$$= -\mathcal{D}_{KL}\left[\mathcal{N}(\mathbf{0}, \sigma_q \mathbb{1})||\mathcal{N}(\mathbf{0}, \mathbb{1})\right] + C - \min_\mathbf{z} \frac{1}{2\sigma^2}\sum_i \min_{\delta_i}\left([\mathbf{G}_c(\mathbf{z}) - \mathbf{x}]_i - \delta_i\right)^2,$$

$$= -\mathcal{D}_{KL}\left[\mathcal{N}(\mathbf{0}, \sigma_q \mathbb{1})||\mathcal{N}(\mathbf{0}, \mathbb{1})\right] + C$$

$$- \min_\mathbf{z} \frac{1}{2\sigma^2}\sum_i \begin{cases} 0 & \text{if } |[\mathbf{G}_y(\mathbf{z}_\mathbf{x}^*) - \mathbf{x}]_i| \leq \epsilon \\ ([\mathbf{G}_y(\mathbf{z}_\mathbf{x}^*) - \mathbf{x}]_i - \epsilon)^2 & \text{if } [\mathbf{G}_y(\mathbf{z}_\mathbf{x}^*) - \mathbf{x}]_i > \epsilon \\ ([\mathbf{G}_y(\mathbf{z}_\mathbf{x}^*) - \mathbf{x}]_i + \epsilon)^2 & \text{if } [\mathbf{G}_y(\mathbf{z}_\mathbf{x}^*) - \mathbf{x}]_i < \epsilon \end{cases}. \tag{14}$$

In this case there is no closed-form solution for the minimization problem on the RHS (in terms of the minimum of $\|\mathbf{G}_c(\mathbf{z}) - \mathbf{x}\|_2$) but we can still compute the solution for each given $\epsilon$ which allows us perform a line search along $\epsilon$ to find the point where equation 13 = equation 14.

## A.7 MODEL & TRAINING DETAILS

**Hyperparameters and training details for the ABS model**  The binary ABS and ABS have the same weights and architecture: The encoder has 4 layers with kernel sizes$= [5, 4, 3, 5]$, strides$= [1, 2, 2, 1]$ and feature map sizes$= [32, 32, 64, 2*8]$. The first 3 layers have ELU activation functions (Clevert et al., 2015), the last layer is linear. All except the last layer use Batch Normalization (Ioffe & Szegedy, 2015). The Decoder architecture has also 4 layers with kernel sizes$= [4, 5, 5, 3]$, strides$= [1, 2, 2, 1]$ and feature map sizes$= [32, 16, 16, 1]$. The first 3 layers have ELU activation functions, the last layer has a sigmoid activation function, and all layers except the last one use Batch Normalization.

We trained the VAEs with the Adam optimizer (Kingma & Ba, 2014). We tuned the dimension $L$ of the latent space of the class-conditional VAEs (ending up with $L = 8$) to achieve 99% test error; started with a high weight for the KL-divergence term at the beginning of training (which was gradually decreased from a factor of 10 to 1 over 50 epochs); estimated the weighting $\boldsymbol{\gamma} = [1, 0.96, 1.001, 1.06, 0.98, 0.96, 1.03, 1, 1, 1]$ of the lower bound via a line search on the training accuracy. The parameters maximizing the test cross entropy[3] and providing a median confidence of $p(y|x) = 0.9$ for our modified softmax (equation 8) are $\eta = 0.000039$ and $\alpha = 440$. For our latent prior, we chose $\sigma_q = 1$ and for the posterior width we choose $\sigma = 1/\sqrt{2}$

**Hyperparameters for the CNNs**  The CNN and Binary CNN share the same architecture but have different weights. The architecture has kernel sizes $= [5, 4, 3, 5]$, strides $= [1, 2, 2, 1]$, and feature map sizes $= [20, 70, 256, 10]$. All layers use ELU activation functions and all layers except the last one apply Batch Normalization. The CNNs are both trained on the cross entropy loss with the Adam optimizer (Kingma & Ba, 2014). The parameters maximizing the test cross entropy and providing a median confidence of $p(y|x) = 0.9$ of the CNN for our modified softmax (equation 8) are $\eta = 143900$ and $\alpha = 1$.

---

[3]Note that this solely scales the probabilities and does not change the classification accuracy.

**Hyperparameters for Madry et al.**     We adapted the pre-trained model provided by Madry et al[4]. Basically the architecture contains two convolutional, two pooling and two fully connected layers. The network is trained on clean and adversarial examples minimizing the cross cross-entropy loss. The parameters maximizing the test cross entropy and providing a median confidence of $p(y|x) = 0.9$ for our modified softmax (equation 8) are $\eta = 60$ and $\alpha = 1$.

**Hyperparameters for the Nearest Neighbour classifier**    For a comparison with neural networks, we imitate logits by replacing them with the negative minimal distance between the input and all samples within each class. The parameters maximizing the test cross entropy and providing a median confidence of $p(y|x) = 0.9$ for our modified softmax (equation 8) are $\eta = 0.000000000004$ and $\alpha = 5$.

---

[4]https://github.com/MadryLab/mnist_challenge

