# OpenReview forum: "Towards the first adversarially robust neural network model on MNIST"
_ICLR.cc/2019/Conference_

### Official Review · AnonReviewer3 · 2018-11-02

**Rating:** 6
**Confidence:** 3

**Review:**

Paper summary: The paper presents a robust Analysis by Synthesis classification model that uses the input distribution within each class to achieve high accuracy and robustness against adversarial perturbations. The architecture involves training VAEs for each class to learn p(x|y) and performing exact inference during evaluation. The authors show that ABS and binary ABS outperform other models in terms of robustness for L2, Linf and L0 attacks respectively.

The paper in general is well written and clear, and the approach of using generative methods such as VAE for better robustness is good.

Pros:
Using VAEs for modeling class conditional distributions for data is an exhaustive approach. The authors show in Fig 4 that ABS generates adversarials that are semantically meaningful for humans, which is not achieved by Madry et al and other models.

Cons:
1) The main concern with this work is that it is heavily tailored towards MNIST and the authors do mention this. Scaling this to other datasets does not seem easy.
2) Using VAEs to model the conditional class distributions is a nice idea, but how does this scale for datasets with large number of classes like imagenet? This would result in having 1000s of VAEs.
3) It would be nice to see this model behaves for skewed datasets.

---

> ### Author Response · Authors · 2018-11-24
> **Scaling to other datasets, skewed datasets**
>
> "The main concern with this work is that it is heavily tailored towards MNIST and the authors do mention this. Scaling this to other datasets does not seem easy. "
> "Using VAEs to model the conditional class distributions is a nice idea, but how does this scale for datasets with large number of classes like imagenet? This would result in having 1000s of VAEs."
>
> First experiments suggest that our robustness is not limited to MNIST. To show this, we trained the proposed ABS model and a vanilla CNN on two class CIFAR and achieve a robustness ~3x larger than a CNN.
>
> Robustness results on 2 class CIFAR:
> model                                   accuracy    |         L2 robustness
> CNN                                       97.1%         |              0.8             (estimated with BIM)
> ABS                                        89.7%         |              2.5            (estimated with LatentDescent attack)
>
> To tackle the reduced accuracy of ABS on CIFAR-10 and other datasets, we are currently working on extensions of our architecture and the training procedure. First experiments show that this can improve the accuracy substantially over baseline ABS and still comes with the same robustness to adversarial perturbations (but this is beyond the scope of this paper).
>
>
> "It would be nice to see this model behaves for skewed datasets."
>
> In contrast to purely discriminative models that require manual rebalancing of the training data, our generative architecture can cope well with unbalanced datasets out of the box. To demonstrate this experimentally, we have trained a two-class MNIST classifier (ones vs. sevens) both on a balanced dataset, an unbalanced datasets (10 times as many sevens than ones during training) and a highly unbalanced dataset (100 times as many ones as sevens during training). They all perform similarly well:
>
>                                                   accuracy      | L_2 median perturbation size with Latent Descent attack
> balanced ABS                       99.6 +- 0.1%   |        3.5  +- 0.1
> 10  :1 unbalanced ABS        99.3 +- 0.2%   |        3.4  +- 0.2
> 100:1 unbalanced ABS        98.5 +- 0.2%   |        3.2  +- 0.2

---

> > ### Public Comment · (anonymous) · 2018-12-06
> > **Question on CIFAR-10 experiments**
> >
> > Why the authors choose these two classes (airplane, automobile) in the experiments, Not all classes on CIFAR 10 ? Do other categories have the similar results?

---

### Official Review · AnonReviewer2 · 2018-11-02
**Nice paper on adversarially robust models**

**Rating:** 7
**Confidence:** 3

**Review:**

In this paper, the authors argued that the current approaches are not robust to adversarial attacks, even for MNIST. They proposed a generative approach for classification, which uses variational autoencoder (VAE) to estimate the class specific feature distribution. Robustness guarantees are derived for their model. Through numeric studies, they demonstrated the performance of their proposal (ABS). They also demonstrated that many of the adversarial examples for their ABS model are actually meaningful to humans, which are different from existing approaches, such as SOTA.

Overall this is a well written paper. The presentation of their methodology is clear, so are the numerical studies.

Some comments:
1) it was not very clear to me that the authors were estimating the p(x) for each y. The transition from p(x|y) to p(x) at the end of page 3 was astute and confused me. The authors should make it more clear.
2) it would be beneficial if the authors could comment on the how strict/loose the lower bound of (2) is, as it is critical in estimating the class specific density.

---

> ### Author Response · Authors · 2018-11-24
> **class conditional probabilities, strictness of ELBO**
>
> "it was not very clear to me that the authors were estimating the p(x) for each y. The transition from p(x|y) to p(x) at the end of page 3 was astute and confused me. The authors should make it more clear."
>
> We agree, thank you for pointing this out. We changed  p(x) -> p(x|y) in Equation (2) and the text.
>
>
> "it would be beneficial if the authors could comment on the how strict/loose the lower bound of (2) is, as it is critical in estimating the class specific density."
>
> For a standard VAE trained on MNIST, the estimate of log p(x) is around -93 while true log-likeilhood is at around -87 (see https://openreview.net/pdf?id=HyZoi-WRb, Figure 3). Hence, the bound is neither extremely loose, nor extremely tight. In any case, one should keep in mind that the goal of the model is not optimal density estimation but accuracy and model robustness, so we can accept to be non-optimal. You may be right, however, that tighter bounds might also increase accuracy and robustness, which is an exciting question to be answered in future work.

---

### Official Review · AnonReviewer1 · 2018-11-03
**a nice paper with space of improvement**

**Rating:** 7
**Confidence:** 4

**Review:**

This paper shows that the problem of defending MNIST is still unsuccessful. It hereby proposes a model that is robust by design specifically for the MNIST classification task. Unlike conventional classifiers, the proposal learns a class-dependent data distribution using VAEs, and conducts variational inference by optimizing over the latent space to estimate the classification logits.

Some extensive experiments verify the model robustness with respect to different distance measure, with most state-of-the-art attacking schemes, and compared against several baselines. The added experiments with rotation and translation further consolidate the value of the work.

Overall I think this is a nice paper. Although being lack of some good intuition, the proposed model indeed show superior robustness to previous defending approaches. Also, the model has some other benefits that are shown in Figure 3 and 4. The results show that the model has indeed learned the data distribution rather than roughly determining the decision boundary of the input space as most existing models do.


However, I have the following comments that might help to improve the paper:

1. It would be more interesting to add more intuition on why the proposed model is already robust by design.

2. Although the paper is designed for MNIST specifically, the proposed scheme should apply to other classification tasks. Have you tried the models on other datasets like CIFAR10/100? It would be interesting to see whether the proposal would work for more complicated tasks. When the training data for each label is unbalanced, namely, some class has very few samples, would you expect the model to fail?

3. Equation (8) is complicated and still model-dependent. Without further relaxation and simplification, it’s not easy to see if this value is small or large, or to understand what kind of message this section is trying to pass.

4. Although the main contribution of the paper is to propose a model that is robust without further defending, the proposed model could still benefit from adversarial training. Have you tried to retrain your model using the adversarial examples you have got and see if it helps?

---

> ### Author Response · Authors · 2018-11-24
> **Scaling to other datasets, unbalanced data, intuition behind the model, robustness lower bound, adversarial training**
>
> "Although the paper is designed for MNIST specifically, the proposed scheme should apply to other classification tasks. Have you tried the models on other datasets like CIFAR10/100? It would be interesting to see whether the proposal would work for more complicated tasks."
>
> First experiments suggest that our robustness is not limited to MNIST. To show this, we trained the proposed ABS model and a vanilla CNN on two class CIFAR and achieve a robustness ~3x larger than a CNN.
>
> Robustness results on 2 class CIFAR:
> model                                   accuracy     |         L2 robustness
> CNN                                       97.1%         |              0.8             (estimated with BIM)
> ABS                                        89.7%          |              2.5            (estimated with LatentDescent attack)
>
> To tackle the reduced accuracy of ABS on CIFAR-10 and other datasets, we are currently working on extensions of our architecture and the training procedure. First experiments show that this can improve the accuracy substantially over baseline ABS and still comes with the same robustness to adversarial perturbations (but this is beyond the scope of this paper).
>
>
> "When the training data for each label is unbalanced, namely, some class has very few samples, would you expect the model to fail?"
>
> In contrast to purely discriminative models that require manual rebalancing of the training data, our generative architecture can cope well with unbalanced datasets out of the box. To demonstrate this experimentally, we have trained a two-class MNIST classifier (ones vs. sevens) both on a balanced dataset, an unbalanced datasets (10 times as many sevens than ones during training) and a highly unbalanced dataset (100 times as many ones as sevens during training). They all perform similarly well:
>
>                                                   accuracy      | L_2 median perturbation size with Latent Descent attack
> balanced ABS                      99.6 +- 0.1%    |        3.5  +- 0.1
> 10  :1 unbalanced ABS       99.3 +- 0.2%    |        3.4  +- 0.2
> 100:1 unbalanced ABS       98.5 +- 0.2%    |        3.2  +- 0.2
>
>
> "It would be more interesting to add more intuition on why the proposed model is already robust by design."
>
> Adversarial training is used to prevent small changes in the input to make large changes in the model decision. In the ABS model, the Gaussian posterior in the reconstruction term ensures that small changes in the input can only entail small changes to the posterior likelihood and thus to the model decision. In other words, small changes in the input can only lead to small changes in the reconstruction error and so the logits (= reconstruction error + KL divergence) can only change slowly with varying inputs.
>
>
> "Equation (8) is complicated and still model-dependent. Without further relaxation and simplification, it’s not easy to see if this value is small or large, or to understand what kind of message this section is trying to pass."
>
> We provide quantitative values in the results section "Lower bounds on Robustness" (we'll add a pointer). For ABS, the mean L2 perturbation (i.e. the mean of epsilon in eq. 8 across samples) is 0.69. For comparison, Hein et al. [1] reaches 0.48.
>
> [1] Matthias Hein and Maksym Andriushchenko. Formal guarantees on the robustness of a classifier against adversarial manipulation. In Advances in Neural Information Processing Systems 30, pp. 2266–2276. Curran Associates, Inc., 2017.
>
>
> "Although the main contribution of the paper is to propose a model that is robust without further defending, the proposed model could still benefit from adversarial training. Have you tried to retrain your model using the adversarial examples you have got and see if it helps?"
>
> It's an interesting question as to whether a combination of analysis by synthesis and adversarial training can yield even better results. One potential problem could be that adversarial training makes little sense if adversarials are already at the perceptual boundary between two classes. This would need to be evaluated carefully and we feel that such an analysis goes beyond the scope of this paper. We will, however, release the code and the pretrained model for the community to play around with such ideas. Thanks for the suggestion!

---

> > ### Public Comment · (anonymous) · 2018-12-06
> > **Have you tried the models on Fashion-MNIST?**
> >
> > As we know,  MNIST is too easy and over used, more importantly, it can not represent modern CV tasks.  Fashion-MNIST is an alternative dataset for MNIST. It would be interesting to see whether the proposal would work for more complicated tasks like Fashion-MNIST.

---

> > > ### Public Comment · (anonymous) · 2018-12-06
> > > **Concur**
> > >
> > > I concur. Fashion-MNIST is a necessary datasets, which is similar to MNIST.  Why not to choose Fashion-MNIST  for analysis. The fact that the method performs well on MNIST is nice, but MNIST should be considered for what it is: a toy dataset.

---

> > ### Public Comment · (anonymous) · 2018-12-06
> > **Robustness results on 3-10 class CIFAR？**
> >
> > Nice to see the proposal work for 2 class CIFAR.  However, as the result show, the accuracy is greatly reduced.
> > There are some questions：
> >
> > 1. Can you show us robustness results on 3-10 class CIFAR? As the category increases, will the robustness decrease?
> >
> > 2. There are 10 categories in CIFAR10,  which two categories did you choose to experiment with? why?
> >
> > 3. What about L0 robustness and L-inf robustness on 2 class CIFAR?

---

### Public Comment · (anonymous) · 2018-10-02
**Related work**

Since a major claim of this paper (the first claim listed in the abstract) is that the Madry et al 2017 model doesn't defend against L0 or L2 attacks, it seems like it would make sense to discuss earlier related work that showed the Madry et al 2017 model doesn't defend against attacks other than Linf threat model it was designed for. To the best of my knowledge, the first such work was the demonstration that it doesn't defend against L1 attacks, which seem to not be mentioned at all in this submission: https://arxiv.org/abs/1710.10733
 There is also the background pixel attack (fig 6 of https://arxiv.org/pdf/1807.06732.pdf ) and a variety of threat models described by https://arxiv.org/abs/1804.03308 where weight decay outperforms Linf-adversarial training.

---

> ### Author Response · Authors · 2018-10-06
> **ABS model is robust to background pixel attack**
>
> Thanks for your comment! We tested our ABS model against one of the background pixel attacks suggested in fig. 6 of https://arxiv.org/pdf/1807.06732.pdf (random lines added on top of the samples) and found a strong robustness against such perturbations (96% accuracy for two lines, 86% for four lines and 54% for eight lines [difficult even for humans], see https://ibb.co/cpDt9K for samples). The combination of Madry et al. with weight decay is certainly interesting but out of the scope of this paper. Thanks for the L1 reference, we'll include it in the manuscript.

---

### Public Comment · ~Florian_Tramer1 · 2018-10-15
**Rotations/translations**

Did you measure the robustness of your model to small (worst-case) rotations and translations? (https://arxiv.org/abs/1712.02779)

I think these attacks could be good candidates to further show that your model is not subject to some form of gradient masking, as the worst-case perturbation can be found via exhaustive search.

Incidentally, rotations and translations are another class of perturbations that the l-infinity model of Madry et al. is not robust against (that's what the above paper by the same authors shows). The paper also shows that you can adversarially train a model to be robust to rotations and translations, but I don't think it says anything about training a model that is robust to both rotations/translations and l-infinity attacks (which your model might be)

---

> ### Author Response · Authors · 2018-10-24
> **Results of spatial transformation attack**
>
> Dear Florian, that's a great suggestion! I took the time to re-implement the spatial attack in Foolbox (because our whole evaluation setup is based on it) and tested (1) a vanilla MNIST network (the one used by Madry et al, as taken from Madry's challenge), (2) the Madry et al. defense (the secret model in Madry's challenge) and (3) our AbS model. We used the same transformation ranges as [Engstrom et al.] (translations: +- 3px, rotation +- 30 degrees). Here are the results:
>
> (1 - Vanilla) Translation-only: 12,3%  ---  Rotation-only: 12.7%  ---  Translation & Rotation: 0.01%
> (2 - Madry) Translation-only:       9%  ---  Rotation-only: 66.0%  ---  Translation & Rotation: 0%
> (3 - AbS)      Translation-only: 25.5%  ---  Rotation-only: 67.1%  ---  Translation & Rotation: 0.3%
>
> I am not yet able to reproduce the large difference between vanilla and defended network present in [Engstrom et al]. We found the defense by Madry et al. work a little worse than reported in [Engstrom et al], in particular with respect to translations, while we found the vanilla network to perform much worse (we used a different one than in [Engstrom et al.] though, which probably explains the difference). AbS performs much better than vanilla in both rotation and translation and also performs better than Madry et al. on shifts. Frankly, I'd expected the AbS to perform even better but on the other hand, if the transformations go beyond the typical transformations of the data than there is no reason why the AbS should learn them.

---

> > ### Public Comment · ~Florian_Tramer1 · 2018-11-02
> > **Transformations beyond the data**
> >
> > Thanks a lot for doing this, very cool results!
> >
> > I agree with your point about the hardness of learning transformations that go beyond what is in the dataset.
> > This raises an interesting question regarding the difference between rotation/translations and l_p perturbations. Intuitively, large l_infty perturbations also go beyond typical data transformations. Yet AbS seems to do fine with them.

---

> > > ### Author Response · Authors · 2018-11-02
> > > **Transformations beyond the data**
> > >
> > > You can think of AbS as incorporating an explicit Gaussian noise model (by means of the Gaussian posterior): it basically assumes that the signal (the digit) is corrupted by noise. In return, as long as the corrupted images stay close (in terms of L2) to the original image, the AbS will not change it's decision. The difference between rotations and L_infty perturbations is that the latter still stay close to the original image in terms of L2 (at least roughly), whereas small rotations can easily lead to large L2 distances.

---

> > > > ### Public Comment · ~Florian_Tramer1 · 2018-11-03
> > > > **Thanks**
> > > >
> > > > That makes sense. Thanks a lot for the explanation.

---

### Author Response · Authors · 2018-10-24
**Small update of L0 results**

Dear reviewers and readers,

we performed additional robustness evaluations and discovered a minor issue with the random seed in the Salt and Pepper (S&P) attack. We reevaluated robustness against S&P as well as Pointwise attack (which uses S&P for initialization) and found small changes in the L0 results:

Format:           Binary ABS robustness  |  ABS robustness

L2 Pointwise Attack:           no change  |    4.8 -> 4.6
L2 overall:                             no change  |     no change

L0 Salt&Pepper Noise:  158.5 -> 146.0  |  182.5 -> 165.0
L0 Pointwise Attack:         36.5 ->  22.0  |   22.0 ->  16.5
L0 overall:                           36.0 ->  21.5  |   22.0 ->  16.5

We will update table 1 and figure 2 in the manuscript accordingly. No conclusions or statements in the paper are affected.

---

### Public Comment · (anonymous) · 2018-11-15
**some questions**



- How is \sigma chosen in Eq.3? Is it different from \sigma_q in Eq.7?

- Why does it make sense to equate (7) and (6), upper and lower bounds? (I'm sure the authors thought it through, but it seems unclear from the text)

---

> ### Author Response · Authors · 2018-11-21
> **sigma in Eq. 3, derivation of estimation of bound for adversarial examples**
>
> - How is \sigma chosen in Eq.3? Is it different from \sigma_q in Eq.7?
>
> The \sigma in Eq. 3 should be called \sigma_q as well, thanks for pointing this out. We set \sigma_q = 1 (the exact value doesn’t really matter at this point since we do not sample from the posterior distribution during the optimization step). We’ll add this to the “Model and Training Details” section in the appendix.
>
>
> - Why does it make sense to equate (7) and (6), upper and lower bounds? (I'm sure the authors thought it through, but it seems unclear from the text)
>
> Remember that an (untargeted) adversarial perturbation tries to maximally lower the likelihood of the true label and to maximally increase the likelihood of some other label. We here derive how much the likelihood of the true label can maximally decrease for a given norm-ball of epsilon (that’s the lower bound), and what the maximum likelihood of any other class may be under the same constraint (that’s the upper bound). The epsilon for which the lower and upper bound are the same is the maximum epsilon for which we can guarantee that the model will still predict the true label.

---

> > ### Public Comment · (anonymous) · 2018-11-23
> > **suggestions**
> >
> > Thanks!
> >
> > The method goes to great computational expense to use Eq. 3 instead of Eq. 2. (8000 evaluations per sample) It would be interesting to see if it's worth it.
> >
> > Also, what if Madry's defense were trained to defend against L2=1.5 attacks? (This seems like a trivial generalization, but I'm not aware of anyone having done this) It would be interesting to see where such a defense would fit in Table 1.

---

> > > ### Author Response · Authors · 2018-11-29
> > > **Inferring latents decoder vs encoder, adversarial training**
> > >
> > > 1. "The method goes to great computational expense to use Eq. 3 instead of Eq. 2. (8000 evaluations per sample) It would be interesting to see if it's worth it."
> > >
> > > We performed a quick experiment with L2 Basic Iterative Method [1] for the ABS model and found that the median L2 robustness with the variational inference (Eq. 2) is 0.05 compared to 2.3 with optimization-based inference (Eq. 3). So indeed the optimization step is crucial to make the model robust.
> > >
> > > [1] https://foolbox.readthedocs.io/en/latest/modules/attacks/gradient.html#foolbox.attacks.L2BasicIterativeAttack
> > >
> > >
> > > 2. "Also, what if Madry's defense were trained to defend against L2=1.5 attacks? (This seems like a trivial generalization, but I'm not aware of anyone having done this). It would be interesting to see where such a defense would fit in Table 1."
> > >
> > > That’s indeed a very interesting question but the generalisation of Madry’s defense is not as trivial (for a fair comparison we have to be careful in choosing the right iterative method (alternative to PGD) and choosing the optimal hyperparameters). We will try to follow up on this in the future but are currently concentrating on other parts of the ABS model (in particular scaling to more complex data sets).

---

### Author Response · Authors · 2018-11-24
**Summary of the updates to the manuscript**

We would like to thank all reviewers for their valuable feedback. Regarding concerns we responded to each reviewer individually

We have uploaded an updated version of the paper with the following changes:

1.) We provide additional intuitions behind the model architecture and its robustness

2.) We have extended the section describing ideas to scale this approach to more complex datasets

3) We provide preliminary results for two class CIFAR.

4) Minor changes
* fixed the correct image for distal adversarials for the ABS model
* We changed p(x) to p(x|y) to be consistent
* We added a pointer to to the results in section 4 "TIGHT ESTIMATES  OF THE LOWER BOUND  FOR ADVERSARIAL EXAMPLES"
* We consistently refer to the sigma of the variational inference as \sigma_q

---

### Public Comment · ~Tianhang_Zheng1 · 2018-11-27
**Interesting method**

A very interesting method! Just two small questions:

1.  As far as I can see, the defense method is quite like a composition of Defense-GAN and binarization method. As claimed in "ensemble adversarial training" (appendix), binarization can help MNIST model robust to Linfinity perturbation. But it is still not very intuitive to me why the model can be robust to L2 perturbation. (The bound given in eq. 8 seems close to 0, does it really make much sense?)

2. Another question is that Defense-GAN can be further attacked by BPDA proposed in https://arxiv.org/pdf/1802.00420.pdf. I was wondering did the proposed method suffer from the same problem (i.e., obfuscated gradients)?

Anyway, the paper is pleasant to read. I would appreciate it if the authors can answer my questions.

---

> ### Author Response · Authors · 2018-11-29
> **DenfenseGAN, L2 robustness intuition, verified lower bound, BPDA**
>
> "1.  As far as I can see, the defense method is quite like a composition of Defense-GAN and binarization method."
>
> Our method is very different from Defense GAN which is basically a sophisticated image denoising followed by a feedforward classifier. In contrast, we use class conditional generative models and no (vulnerable) feedforward classifier at all.
>
>
> "As claimed in "ensemble adversarial training" (appendix), binarization can help MNIST model robust to Linfinity perturbation. But it is still not very intuitive to me why the model can be robust to L2 perturbation.
>
> Our intuition behind the ABS models L2 robustness is due to the Gaussian posterior (in pixelspace) in the reconstruction term, which ensures that small changes in the input can only entail small changes to the posterior likelihood and thus to the model decision. In other words, small changes in the input can only lead to small changes in the reconstruction error and so the logits (= reconstruction error + KL divergence) can only change slowly with varying inputs.
>
>
> "(The bound given in eq. 8 seems close to 0, does it really make much sense?)"
>
> Our verified lower bound of the mean L_2 robustness is 0.690 ± 0.005 which is quite high compared to other SOTA methods which provide guarantees for the lower bound (i.e. Hein et al. [1] who have 0.48).
>
> [1] Matthias Hein and Maksym Andriushchenko. "Formal guarantees on the robustness of a classifier against adversarial manipulation". In Advances in Neural Information Processing Systems 30, pp. 2266–2276. Curran Associates, Inc., 2017.
>
>
> "2. Another question is that Defense-GAN can be further attacked by BPDA proposed in https://arxiv.org/pdf/1802.00420.pdf. I was wondering did the proposed method suffer from the same problem (i.e., obfuscated gradients)?"
>
> BPDA is not really an attack but a way to recover proper gradients for certain models (e.g. by pass-through estimators [2]). We do this in two ways: first by computing descent directions in the low-dimensional latent space (LatentDescentAttack) and second by estimating gradients using a finite-difference estimate (+ a "pass through estimator" [2] for the binary ABS model). The LatentDescentAttack is closest in spirit to BPDA (but adapted to our model).
>
> [2] Bengio, Y., Leonard, N., and Courville, A. "Estimating or propagating gradients through stochastic neurons for conditional computation". arXiv preprint arXiv:1308.3432, 2013.

---

> > ### Public Comment · ~Tianhang_Zheng1 · 2018-11-29
> > **Thanks for your detailed reply**
> >
> > Thanks for your detailed reply, which clarifies most of my minor questions and concerns.
> >  Well, defense-GAN also learn a z in the latent space iteratively such that G(z) is close to the sample x. That's why I thought the methodology used in defense-GAN is somehow similar to what you did in the optimization-inference step. But defintely I see ABS is different from Defense-GAN. For BPDA, I know it is method to recover gradient, actually more precisely, estimate the non-zero and finite gradient that can be used for gradient-based algorithms. It seems like LatentDescentAttack did the similar thing.
> >  Thanks again, since I got many insights from the paper and your reply.

---

### Public Comment · (anonymous) · 2018-12-07
**attacks mode**

A nice paper, but there is one place I don't know very well.

Whether the attacks is in white box mode or black box mode？

I would appreciate it if the authors can answer my question.

---

> ### Author Response · Authors · 2018-12-08
> **white box**
>
> We assume full knowledge of the model (= white-box setting) which we use to design a customised attack that optimises in the hidden latent space of the model. Furthermore we use score-based and decision-based adversarial attacks.

---

### Meta-Review · Area_Chair1 · 2018-12-15
**Interesting progress on more versatile robustness guarantees**

**Confidence:** 5
**Recommendation:** Accept (Poster)

**Metareview:**

The paper presents a technique of training robust classification models that uses the input distribution within each class to achieve high accuracy and robustness against adversarial perturbations.

Strengths:

- The resulting model offers good robustness guarantees for a wide range of norm-bounded perturbations

- The authors put a lot of care into the robustness evaluation

Weaknesses:

- Some of the "shortcomings" attributed to the previous work seem confusing, as the reported vulnerability corresponds to threat models that the previous work did not made claims about

Overall, this looks like a valuable and interesting contribution.

---

> ### Public Comment · (anonymous) · 2018-12-26
> **Threat models?**
>
> Why is the l-0 norm an interesting threat model? Although it's a precursor to a more useful form of general robustness, it is very limited and it is not interesting to limit an attacker to this threat model. As such, it is indeed a short-coming of that defense that the robustness does not carry over to other threat models.